# Antifungal Capacity of Poolish-Type Sourdough Supplemented with *Lactiplantibacillus plantarum* and Its Aqueous Extracts In Vitro and Bread

**DOI:** 10.3390/antibiotics11121813

**Published:** 2022-12-14

**Authors:** Ricardo H. Hernández-Figueroa, Emma Mani-López, Aurelio López-Malo

**Affiliations:** Departamento de Ingeniería Química, Alimentos y Ambiental, Universidad de las Américas Puebla, San Andrés Cholula 72810, Mexico

**Keywords:** *Lactiplantibacillus plantarum*, sourdough, antifungal activity, poolish-type

## Abstract

This study aimed to evaluate the antifungal capacity of the aqueous extracts (AE) of poolish-type sourdoughs fermented with *Lactiplantibacillus plantarum* NRRL B-4496 on broth, agar, and bread. The aqueous extracts were obtained by centrifugation and separating the supernatant from the poolish sourdoughs once the fermentation time had ended. The aqueous extracts inhibited 80% of the growth of *Penicillium chrysogenum* and *Penicillium corylophilum* and <20% of *Aspergillus niger* in broth. The AEs delayed the radial growth rate and increased the lag time for the three molds tested. The addition of poolish-type sourdoughs inhibited fungal growth in bread for ten days. The extracts’ fungistatic capacity was primarily attributed to lactic and acetic acids and probably the antifungal peptides occurring in the AE. The *L. plantarum* sourdough is an alternative to calcium propionate as an organic antifungal agent.

## 1. Introduction

Nowadays, the demand for organic and synthetic additive-free foods has led to the search for natural antimicrobial alternatives. Regarding bakery products, consumers want bread free of synthetic additives. However, bread has a short shelf life due to the growth of molds [1,2]. Sourdoughs result from the fermentation of cereal flour with lactic acid bacteria (LAB) and/or yeast, commonly used as an ingredient in the manufacture of bread [3]. There are different types of sourdoughs, including poolish-type sourdoughs (Type II) which are the result of the inoculation of a flour-water mixture with different types of LAB and the addition or not of *Saccharomyces cerevisiae*. LAB are microorganisms commonly isolated from sourdoughs and are important due to their organic acid production, among other compounds. The fermentation of these sourdoughs occurs at temperatures above room temperature and typically lasts from one to three days [4]. The use of sourdough in manufacturing bakery products is a possible antimicrobial alternative since, for many years, and traditionally, sourdough has been used in bread making for improving organoleptic properties such as flavor, color, and texture. 

The process of lactic acid fermentation in sourdoughs depends directly on the type of fermenting microorganism. The LAB isolated in these doughs can be classified according to the glycolytic reactions carried out in the glycolysis metabolic pathway as homofermentative and heterofermentative [5]. Some studies have shown that during the fermentation of sourdoughs, heterofermentative LAB produced lactic acid, acetic acid, ethanol, and CO_2_. Moreover, the LAB present in the sourdoughs have proteases, peptidases, and amino acid converting enzymes which break down the proteins of cereals to produce compounds with bioactive properties such as bacteriocins [6]. Different studies have shown that organic acids such as acetic, lactic, and phenyllactic acids have an antimicrobial effect and, in some cases, an antifungal effect [6,7,8].

Cell-free supernatant refers to the transparent liquid obtained after the growth of a specific LAB in a liquid or semi-liquid medium after centrifugation and filtration (0.45 μm) [9]. The aqueous extracts (AEs) from sourdough are the cell-free supernatant liquids resulting from centrifugation and filtration. Furthermore, the AEs from sourdough have shown the ability to decrease the growth of a wide number of molds responsible for the spoilage of bakery products and improve the products’ physicochemical properties [10,11,12]. Previous studies have shown that the AEs from sourdoughs fermented with *Lactobacillus plantarum* CRL 759 could decrease (>40%) the radial growth of *Aspergillus niger* [1]. Similarly, Rizzello et al. [13] observed that AEs obtained from sourdoughs added with wheat germ and fermented with *Lactobacillus plantarum* LB1 and *Lactobacillus rossiae* LB5 delayed the growth of some of the most relevant spoilage fungi in baked products.

The direct use of fermented sourdoughs in bread formulation to slow the growth of spoilage microorganisms has also been investigated. Luz et al. [12] reported that sourdough formulated with whey and additionally fermented with *L. plantarum* CECT 749 or *Lactobacillus bulgaricus* CECT 4005, increased their shelf life by 2 or 4 days, respectively. Likewise, they observed that during sourdough fermentation with both *Lactobacillus* species, different organic acids were formed and related with the ability to retard the growth of various species of *Penicillium*, *Fusarium*, and *Aspergillus*. However, even though sourdoughs and their aqueous extracts have exhibited antimicrobial activity, few studies have investigated their antifungal activity. Therefore, the objective of this work was to evaluate the in vitro and bread antifungal capacity of AEs obtained from poolish-type sourdoughs supplemented with *Lactiplantibacillus plantarum* NRRL B-4496 against *A. niger*, *Penicillium chrysogenum*, and *Penicillium corilophylum*, in addition to determining the physicochemical properties of the bread.

## 2. Materials and Methods

### 2.1. Culture Conditions and Aqueous Extract Preparation

*Lactiplantibacillus plantarum* NRRL B-4496 (formerly *Lactobacillus plantarum* [14]) was acquired in lyophilized form from the USDA (Agricultural Research Service, Peoria, IL, USA). The strain was cultivated in de Man Rogosa Sharpe (MRS) broth (Difco™, BD, Sparks, MD, USA) at 35 °C for 48 h without shaking. Cells were harvested by centrifugation at 7000× *g* for 25 min at 5 °C (Marathon 21K/R, Fischer Scientific, Schwerte, Germany) and used to formulate the poolish-type sourdough. The sourdough was prepared from a standard formulation using refined wheat flour (protein content of 11.1 g/100 g) and water in equal parts (1:1) and 1% (*w*/*w*) of the pellet of *L. plantarum* wet cells. The ingredients were mixed until a homogeneous dough was obtained and fermented at 35 °C for 24 (PD24), 48 (PD48), or 72 (PD72) h. Different fermentation times were tested to identify whether the antifungal activity was dependent on fermentation time. To obtain the AEs, the sourdoughs were centrifuged at 7000× *g* for 25 min at 5 °C and filtered through 0.45 μm cellulose nitrate filter (Advantec, MFS, Dublin, CA, USA). The AEs were stored at −18 °C until their use.

*A. niger*, *P. chrysogenum*, and *P. corilophylum* were obtained from the Food Microbiology Laboratory of the Universidad de las Americas Puebla. Molds were cultured on potato-dextrose agar (PDA; Bioxon, BD, Estado de México, Mexico) slants for 7 d at 25 °C. Then, 5 mL of a sterile solution of Tween 80 (0.4% *w*/*v*) (Sigma-Aldrich, St. Louis, MO, USA) was poured on the surface of each mold culture and the slant was gently shaken for 2 min to favor the harvesting of the spores. A hemocytometer and an optical microscope were used to determine the number of spores in the suspension per milliliter. The number of spores was adjusted to 10^4^ spores/mL with the Tween solution.

### 2.2. Determination of pH, Titratable Acidity, and Reducing Sugars of Aqueous Extracts (AE)

The pH was measured by electrode immersion with a pH meter (HI 2210 Hanna Instruments, Woonsocket, RI, USA) following the AACCI method 02-52 [15]. The total titratable acidity was determined according to the AACCI method 02-31 [15], and the results were expressed as milligrams of lactic acid per ml of sample. The reducing sugars were quantified following the Lane–Eynon method (31.036, AOAC [16]). Measurements were performed in triplicate.

### 2.3. Antifungal Activity of Aqueous Extracts

#### 2.3.1. Mold Inhibition in Liquid Media

The assay was performed as described by Luz et al. [12] with some modifications. A total of 100 µL of malt extract broth and 50 µL of inoculum (1 × 10^4^ spores/mL) were placed into each well of a microplate (96 wells); then, 12 concentrations (0, 5, 9, 14, 19, 23, 28, 33, 37, 42, 46, and 50%) of AEs were added based on 300 µL of the total well volume. Each well was adjusted with sterile water to the final volume. The plates were incubated at 25 °C for 3 d, and OD630 was measured daily with an automatic microplate reader model ELx800 (BioTek Instruments, Winooski, VT, USA) to assess the antifungal activity. Wells without inoculum or AE were also tested as controls. To verify the mold inhibition, 25 µL of each well with an OD ≤ 0.3 was plated on PDA, plates were incubated at 25 °C for 3 d, and survivors were counted. In order to identify the antifungal compounds in the AEs, the AEs were neutralized with a 40% NaOH (*w*/*v*) solution to pH 6.5. Afterward, the antifungal activity was evaluated as described before. Similarly, to determine if the antifungal compound responsible was peptide-like, 10 mL of neutralized AEs were mixed with 20 μg/mL solution of proteinase K (Sigma-Aldrich, St. Louis, MO, USA) and incubated at 37 °C ± 1 °C for 1 h. Following this, the treated AE was heated (121 °C) for an additional 5 min [17]. Afterward, the antifungal activity was evaluated as described before.

#### 2.3.2. Inhibition of Radial Mold Growth

For this test, the AE from the poolish-type sourdough was centrifuged at 7000× *g* for 25 min at 5 °C, and the supernatant was pasteurized (PPD) at 100 °C for 1 min. Sterile melted PDA was mixed with the AE at four concentrations (20, 26, 33, and 50% *v*/*v*). Each agar-AE mixture was poured into Petri dishes and inoculated with 5 µL of spore suspension (1 × 10^6^ spores/mL) in the center of the plate. Colony diameter measurement was performed at three points in the growing colony every 24 h until mold growth reached the total diameter of the plate. The molds’ radial growth was modeled with the modified Gompertz equation (Equation (1)) [18], and the model parameters were fitted by non-linear regression.
(1)$$\ln\frac{D}{D_{0}}=A\times \exp\left[-\exp\left[\left(\frac{\mu \times e}{A}\right)\left(\lambda -t\right)+1\right]\right]\;$$
where *μ* is the maximum growth rate (1/h), *A* is the maximum diameter observed (mm), *λ* is the phase of adaptation or lag time (h), *D* is the colony diameter (mm) at time *t* (h), *D*_0_ is the colony diameter at time 0, and *e* is the “Euler’s number” (exp (1)). To analyze the models’ goodness of fit, residual analysis and correlation coefficients (R^2^) were calculated with Minitab 20 software (Minitab LLC, State College, PA, USA).

### 2.4. Lactic and Acetic Acids Analysis in Aqueous Extracts

The analysis and quantification of the concentration of lactic and acetic acids were carried out by high-performance liquid chromatography (HPLC) using an Agilent 1260 chromatograph (Agilent Technologies, Santa Clara, CA, USA) equipped with a diode-array detector (DAD) programmed to a wavelength of 210 nm. Filtered AEs were sampled with an Agilent G1329 autosampler (Agilent Technologies, Santa Clara, CA, USA) with an injection volume of 20 µL. The separation of compounds was performed in an Aminex HPX-87H column (300 × 7.8 mm) (BIO-RAD, Hercules, CA, USA) using a monobasic potassium phosphate buffer (20 mM) solution (pH 2.4, adjusted with phosphoric acid) as the isocratic mobile phase at 0.6 mL/min at room temperature. Standard solutions at 30–400 mM were prepared to quantify the lactic, acetic, propionic, butyric, and pyroglutamic acids. The peak area for each solution was correlated with the concentration using a linear fit. For the standards, the correlation coefficients (R^2^) were >0.99.

### 2.5. Baking Procedure

Wheat flour 38%, water 23%, yeast 1%, and poolish-type sourdough (formulated with or without *Lactiplantibacillus plantarum* for 48 h) 28% were utilized for the bread formulation. The ingredients were mixed and kneaded using a Legacy HL200 mixer/kneader (Hobart, Troy, OH, USA) for 30 min until a flexible and elastic dough was obtained. Afterward, the dough was divided into pieces (100 g), rounded, and manually formed into loaf-shaped pieces. The formed pieces of bread were placed in the fermenter oven of the Mini combo (Zucchelli Alpha, Trevenzuolo, Verona Italy) for 60 min at 30 °C and 65% relative humidity. Next, the baking process was performed at 200 °C for 18 min in a Mini combo electric oven (Zucchelli Alpha, Trevenzuolo, Verona Italy). Then, the bread was cooled at room temperature for 60 min. Finally, the bread was stored for 14 d at room temperature in sealed polyethylene bags.

### 2.6. Determination of pH, Moisture, and a_w_ of the Bread

For the bread’s pH determination, a pH meter model HI2210 (Hanna Instruments, Woonsocket, RI, USA) was used following the 02-52 method of the AACCI [15]. For moisture and a_w_ tests, the crumb and crust from the loaves of bread were separated and analyzed individually. The moisture content was determined following the AOAC 930.15 [16]. AquaLab 4TEV Series equipment (Meter Food, Pullman, WA, USA) was used for a_w_ analysis. pH measurements were carried out at days 0 and 14 of the storage and a_w_ at days 0, 1, 4, and 14. All determinations were performed in triplicate.

### 2.7. Antifungal Activity in Bread

The packaged loaves of bread were examined daily for 28 days or until mold growth was detected on the surface.

### 2.8. Statistical Analysis

The results were analyzed with the Analysis of Variance (ANOVA) and Tukey’s mean comparison test (*p* < 0.05) using Minitab 20 software (Minitab LLC, State College, PA, USA).

## 3. Results and Discussion

### 3.1. pH, Titratable Acidity, and Reducing Sugars of AE

Table 1 shows the results of pH, titratable acidity (TA), and reducing sugars for the different fermented poolish-type doughs *Lactiplantibacillus plantarum* NRRL B-4496. As the fermentation time increased, the pH declined, and the TA increased due to the formation of organic acids by *Lactiplantibacillus plantarum* NRRL B-4496. Sourdoughs fermented for 48 h (Poolish Dough, PD48h) and 72 h (PD72h) showed the lowest pH values and, therefore, the highest titratable acidity values, which were similar (*p* > 0.05). The control dough without *Lactiplantibacillus plantarum* NRRL B-4496 showed a higher pH. Settanni et al. [19] obtained similar results of pH (4.0–3.7) and TA (1–1.3% *w*/*v*) for sourdoughs inoculated with *L. plantarum* PON 100274 and fermented for 21 h. Similarly, they observed an increase in the TA of sourdoughs which confirms the change in pH was directly related to the production of organic acids, such as lactic acid (1.3–4 mg/g) and acetic acid (0.15–0.80 mg/g), by the *L. plantarum*.

The carbohydrates in wheat flour are starch, maltose, sucrose, glucose, and fructose [20]. During the lactic acid fermentation of sourdough, the enzymes of the LAB, especially hydrolases and phosphohydrolases, begin to separate the maltose from the refined flour generating reducing sugars, which are required for LAB’s metabolic activity [6]. The reducing sugar content of sourdoughs prepared with *Lactiplantibacillus plantarum* NRRL B-4496 (Table 1) confirms the hydrolysis process during fermentation, which was time-dependent. As expected, the control dough presented the lowest concentration of reducing sugars. Similar results were observed by Belz et al. [21] in sourdoughs fermented with *Lactobacillus amylovorus* DSM 19280. The sourdoughs fermented with *L. amylovorus* presented a high amount of reducing sugars (glucose, fructose, and maltose).

### 3.2. Antifungal Activity by OD630nm and Microbial Counts

It was determined from optical density tests using the 12 concentrations of AEs against molds that the concentrations with the highest fungal inhibition were 20, 26, and 33%, which inhibited between 76–98% of mold growth. These three concentrations were used for plate counts. Figure 1 shows the percentages of inhibition for each analyzed mold. As can be seen, the mold that presented the greatest resistance to AEs was *A. niger*, and the inhibition was <20%; the fermentation time and AE concentration did not show significant differences (*p* > 0.05) for this mold. In general, the PD48 and PD72 extracts had a great percentage of inhibition (>85%) against *P. chrysogenum* and *P. corylophilum*, with no significant differences observed compared with calcium propionate (CP). Only AEs at 20% obtained from the PD48 dough presented an inhibition <85% against *P. chrysogenum* and was significantly different (*p* < 0.05) from the 26 or 33% concentrations. In contrast, the PD24 extract showed the lowest inhibition (<80%) and was significantly different (*p* < 0.05) from PD48 and PD72. None of the AEs completely inhibited the growth of the studied molds. Thus, AEs obtained from sourdoughs fermented with *Lactiplantibacillus plantarum* NRRL B-4496 have a fungistatic capability which is in accordance with the studies of Cizeikiene et al. and Gerez et al. [22,23]. Demirbaş et al. [24] found that the cell-free supernatants (CFS) from *L. plantarum* N-15 strongly inhibited the spore formation and micellar growth of *P. chrysogenum*. The antifungal activity of LAB is attributed to the production of organic acids and a low pH that affect the motive force of protons in the cell membrane, leading to decreasing fungal growth [25]. Based on these results, further tests were performed with the PD48 extract.

Figure 2 shows the effect of PD48 extract on mold growth measured by optical density. For *A. niger*, all AE concentrations increased the lag phase. However, after 72 h of incubation, the OD for all concentrations was higher than the control and CP. This phenomenon is probably due to the higher content of reducing sugars in the extract that molds can use as a carbon source. In contrast, the concentration of the antimicrobial compounds was not enough to inhibit mold growth [9].

As expected, all tested concentrations of PD48 AE showed an increase in the lag phase of *P. chrysogenum* and *P. corylophilum*, having a lower OD compared with the control after 72 h (Figure 2). Likewise, the CFS obtained from the fermentation of hydrolyzed wheat flour (WFH) delayed 100% of the growth of *P. corylophilum* [7,9]. In the present work, the AEs did not inhibit 100% of *P. corylophilum* growth, probably owing to unhydrolyzed flour being used for *Lactiplantibacillus plantarum* NRRL B-4496 fermentation which limited the synthesis of metabolites (organic acids and others).

On the other hand, the neutralized AE maintained its inhibitory capability against the tested molds. Figure 3 exhibits the mold inhibition at three concentrations of neutralized AE (PD48N) which are similar (*p* > 0.05) to native AEs for *A. niger* and *P. corylophilum*; only the neutralized AE at 20% (PD48N) on *P. chrysogenum* presented a significant high inhibition (*p* < 0.05) compared with native AE (PD48) at the same concentration. The neutralized AEs treated with proteinase K lost their antifungal capacity against all the molds analyzed. Similar findings were reported by Arrioja-Bretón et al. [17] for CFS obtained from *Lactiplantibacillus plantarum* NRRL B-4496 cultivated in MRS broth, which maintained its antimicrobial activity against pathogenic bacteria after CFS was neutralized at pH 6.5. Moreover, fractionated CFS from *Lactiplantibacillus plantarum* NRRL B-4496 grown in MRS broth showed antimicrobial protein compounds against *Listeria monocytogenes* Scott A [26]; thus, antifungal protein/polypeptide compounds are likely present in the AE. Likewise, Rizzello et al. [13] observed the antifungal activity of methanol and water/salt-soluble extracts obtained from the fermentation of wheat germ with *L. plantarum* LB1 and *L. rossiae* LB5 against various molds isolated from bread. They reported that the antifungal activity was due to the synergistic and complex activity between the organic acids and peptide compounds formed during the lactic acid fermentation of the two lactobacilli. Further research should be performed to identify the antifungal compound in this work.

### 3.3. Inhibition of Radial Mold Growth

Table 2 shows the parameters of the modified Gompertz equation obtained from mold radial growth; in all cases, the fitted of the prediction model was excellent (R^2^ > 0.98) (Figure 4). In general, the growth rate (µ) decreased as the PPD concentration increased; *A. niger* was the most resistant mold to the tested PPD. Figure 4 shows that the growing trend of *A. niger* is very similar between all the PPD concentrations tested. However, the incorporation of 33% of PPD had the highest inhibition against *A. niger*. In addition, the lag phase (λ) (Table 2) slightly increased when 20, 26, or 50% of PPD were used. Similar results were reported by Russo et al. [27] for *A. niger* treated with CFS (12% *v*/*v*) from different strains of *L. plantarum* cultured in MRS broth; the inhibition levels were 13–15%. Likewise, Samapundo et al. [2] observed that the addition of 3% (*w*/*v*) of commercial fermentates obtained from cereal flours significantly decreased (20–25%) the fungal growth of *P. chrysogenum* and *P. paneum*. As expected, calcium propionate had the highest effect on the growth rate; however, it did not extend the lag phase of *A. niger*.

The fungistatic effect of PPD was most effective for *P. chrysogenum* and *P. corylophilum* (Figure 4) when 50% was supplemented. *P. chrysogenum* and *P. corylphilum* growth rates were delayed by ~24% and 21%, respectively, whereas the lag phase was extended for 24 h and 17 h, respectively. Although CP showed good antimicrobial activity against both *Penicillium*, the fungistatic effect of CP was lower compared with PPD at 33% for *P. chrysogenum* and 50% for *P. corylophilum*.

The fungistatic capacity of CFSs is primarily due to the content of organic acids such as lactic acid, acetic acid, and phenyllactic acid, among others, which provoke deformations and the size reduction in hyphae, and in some cases, cause lesions on the mycelium of the mold [9]. However, previous studies have shown that organic acids have weak effects against spore formation and germination for different *Penicillium* species [11,28]. Figure 5 displays the molds’ development on agar supplemented with different levels of PPD or CP for 72 or 192 h of incubation. As mentioned previously, PPD delayed the molds’ growth (colony diameter size) and retarded the spore formation and hyphae development and abundance as a consequence of the antifungal compounds (Figure 5). However, in small mold colonies, spore formation was not inhibited by either the PPD or the CP for the *Penicillium*. For *A. niger*, only CP decreased the spore formation.

### 3.4. Quantification of Lactic and Acetic Acids in Aqueous Extracts

The HPLC analysis revealed the presence of lactic and acetic acid in the AEs (Table 3); as expected, lactic acid was the major in all extracts. The PD48 and PD72 extracts had the largest concentration of lactic acid and acetic acid (*p* < 0.05), followed by the PD24 sample. Gerez et al. [23] reported higher lactic acid concentrations and similar levels of acetic acid in the CFS from various *L. plantarum* strains cultured in MRS broth for 24 h. They recorded lactic acid levels ranging from 216 to 236 mM and acetic acid levels between 13 and 28 mM. Propionic, butyric, and pyroglutamic acids were not detected in the AEs. Organic acid production during lactic acid fermentation depends on the LAB genus, specie, strain, and the available substrates in the growth medium [29]. Thus, different levels in organic acid production were expected compared with previous reports.

### 3.5. Antifungal Activity of Poolish-Type Sourdough Fermented with Lactiplantibacillus plantarum on Bread

Table 4 presents the mold growth on bread loaves during storage at 25 °C. After 5 d of storage, more than 80% of the control breads showed fungal growth, whereas breads supplemented with PD48h sourdough (BPD48h) were mold-free. In total, 50% of the poolish-type sourdough breads supplemented with *Lactiplantibacillus plantarum* NRRL B-4496 exhibited mold growth after 10 or 14 d of storage. Figure 6 displays the control pieces of bread and those formulated with poolish-type sourdough during storage. The addition of PD48h sourdough into the bread formulation extended the shelf life by 6 d compared with the control. Similar results were reported by Luz et al. [12] when sourdough fermented with *L. plantarum* CECT749 added into bread formulation prolonged the bread’s shelf life for at least 2 d.

Concerns about the safety of the use of *Lactiplantibacillus plantarum* NRRL B-4496 in food production could limit its use. Despite this, studies on human consumption of the specific strain (NRRL B-4496) have not been reported; this bacterium has a millenary tradition of safe use in fermented foods [30]. In addition, *Lactiplantibacillus plantarum* is recognized under a Qualified Presumption of Safety by the EFSA BIOHAZ Panel [31] since the updated division of the genus *Lactobacillus*. Therefore, studies on the human biological effects of derived AEs, pasteurized AEs, and poolish-type sourdough from *Lactiplantibacillus plantarum* are encouraged to confirm their safe use.

### 3.6. Bread pH, Moisture Content, and a_w_ during the Storage

The pH of the control bread was 6.25 (crust) and 6.20 (crumb), while the bread with BPD48h at time zero had a pH of 5.14 (crust) and 5.16 (crumb); these values were maintained during the storage. Figure 7 shows the moisture content values and a_w_ of the crust and crumb of control and PD48h loaves of bread during storage. After baking, the control and PD48h breads had a similar moisture content (crumb and crust), whereas a_w_ was lower for PD48h bread (crumb and crust). In stored bakery products, a mass transfer phenomenon occurs since water moves from the crumb to the crust until the moisture content is very similar [32]. This behavior was observed for control and PD48h breads (Figure 7), and the moisture content difference between the crumb and crust of both breads was similar (~3 g H_2_O/g sample) after 14 d. Furthermore, the a_w_ values of the crumb and crust of the control and the bread supplemented with BPD48h after 14 d of storage were similar and ranged from 0.8904 to 0.9097, which were favorable for fungal growth (0.88–0.80) [28,33]. Therefore, the mold growth restriction on bread was only attributed to PD48h sourdough.

## 4. Conclusions

The aqueous extracts from poolish-type sourdough fermented with *Lactiplantibacillus plantarum* NRRL B-4496 presented important antifungal activity against *Penicillium corylophilum* and *P. chrysogenum*. The antifungal activity is related primarily to the organic acids (lactic and acetic); however, a protein/polypeptide compound may also be involved. Furthermore, the supplementation of poolish-type sourdough fermented with *L. plantarum* into a bread formulation significantly increased the bread´s shelf life since mold growth was inhibited. Thus, the studied poolish-type sourdough could be an organic antifungal alternative to the synthetic ones in bakery products. Although the long-traditional use of *Lactiplantibacillus plantarum* has been recognized in fermented foods, studies on specific strains with regard to the effects of AEs, pasteurized AEs, and poolish-type sourdough in humans are necessary.

## Figures and Tables

**Figure 1 antibiotics-11-01813-f001:**
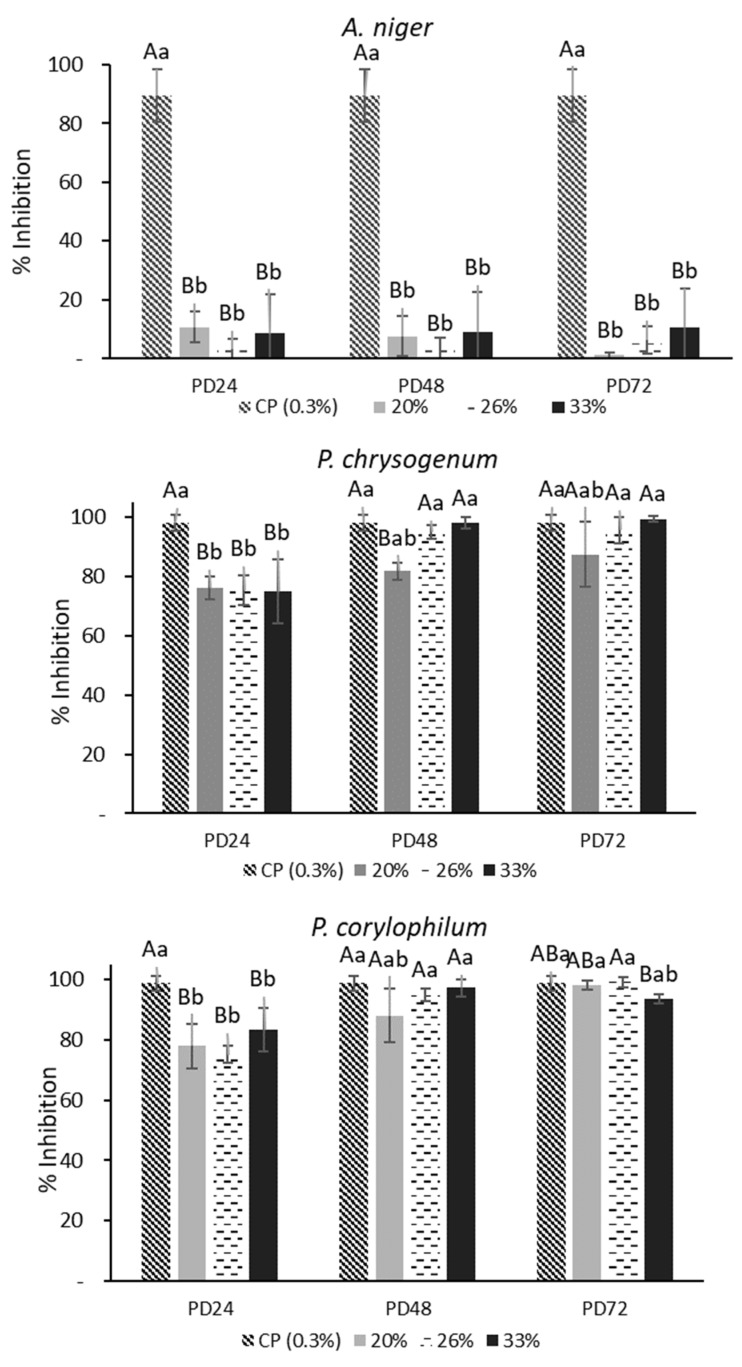
Percentage of inhibition (plate counts) of aqueous extracts (AEs) obtained from the poolish-type dough (PD) fermented with *Lactiplantibacillus plantarum* NRRL B-4496 for 24, 48, and 72 h at three concentrations and calcium propionate (CP). Different capital letters show significant differences (*p* < 0.05) with respect to the fermentation time. Lowercase letters indicate significant differences (*p* < 0.05) with respect to the concentration of AEs at the same fermentation time.

**Figure 2 antibiotics-11-01813-f002:**
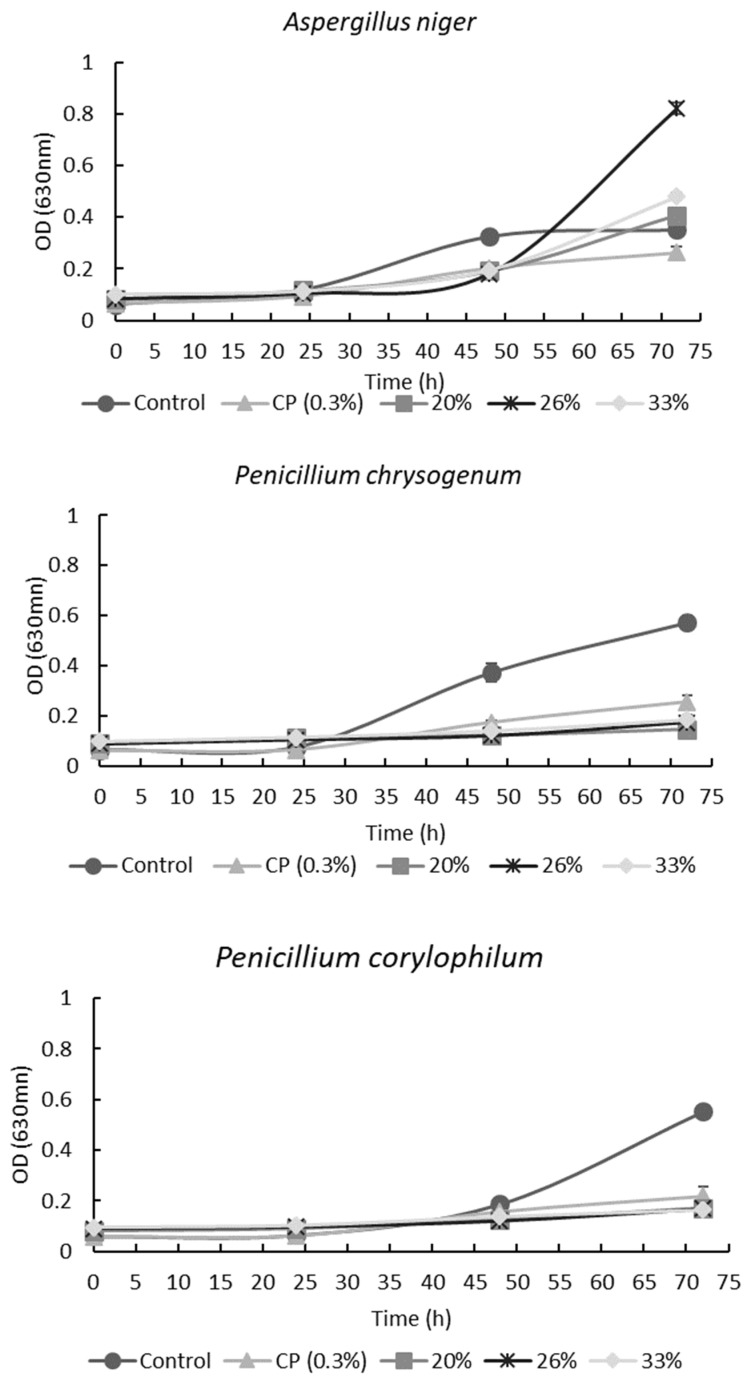
Effect of different concentrations of aqueous extracts from poolish-type sourdough fermented with *Lactiplantibacillus plantarum* NRRL B-4496 at 48 h and calcium propionate (CP) on molds’ growth by optical density at 630 nm.

**Figure 3 antibiotics-11-01813-f003:**
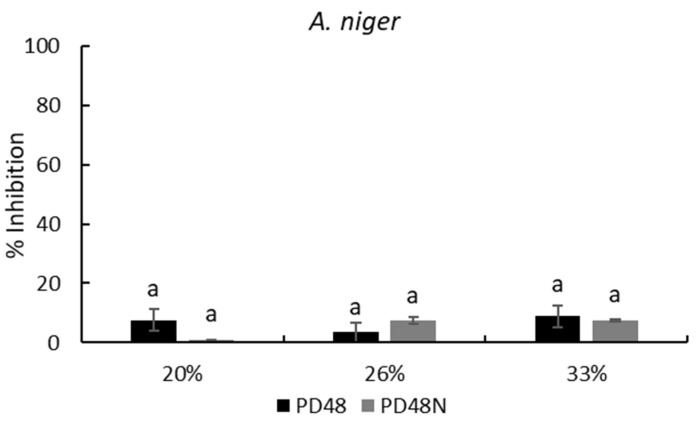
Percentage of inhibition (plate counts) of native (PD48) or neutralized (PD48N) aqueous extracts (AEs) obtained from the sourdough fermented with *Lactiplantibacillus plantarum* NRRL B-4496 after 48 h at three concentrations. Different lowercase letters indicate significant differences (*p* < 0.05) with respect to native or neutralized AEs.

**Figure 4 antibiotics-11-01813-f004:**
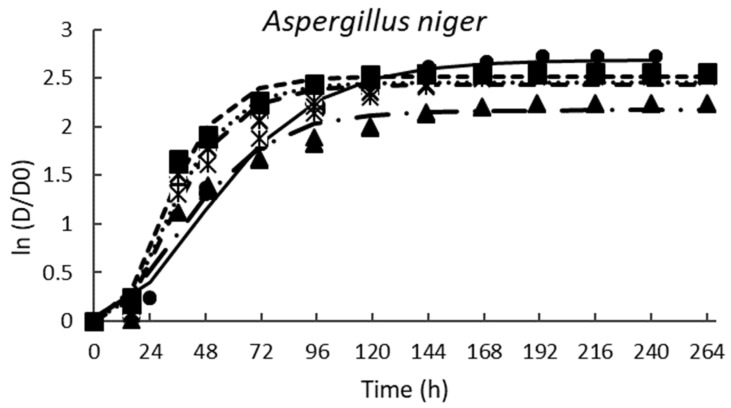
Effect of the different concentrations of pasteurized aqueous extracts (AE) from poolish-type sourdough fermented with *Lactiplantibacillus plantarum* NRRL B-4496 for 48 h on the growth of the tested molds. Potato-dextrose agar (PDA) control (◾), PDA + 20% AE (♦), PDA + 26% AE (*), PDA + 33% AE (▲), PDA + 50% AE (•), and lines are the predicted values using the modified Gompertz equation.

**Figure 5 antibiotics-11-01813-f005:**
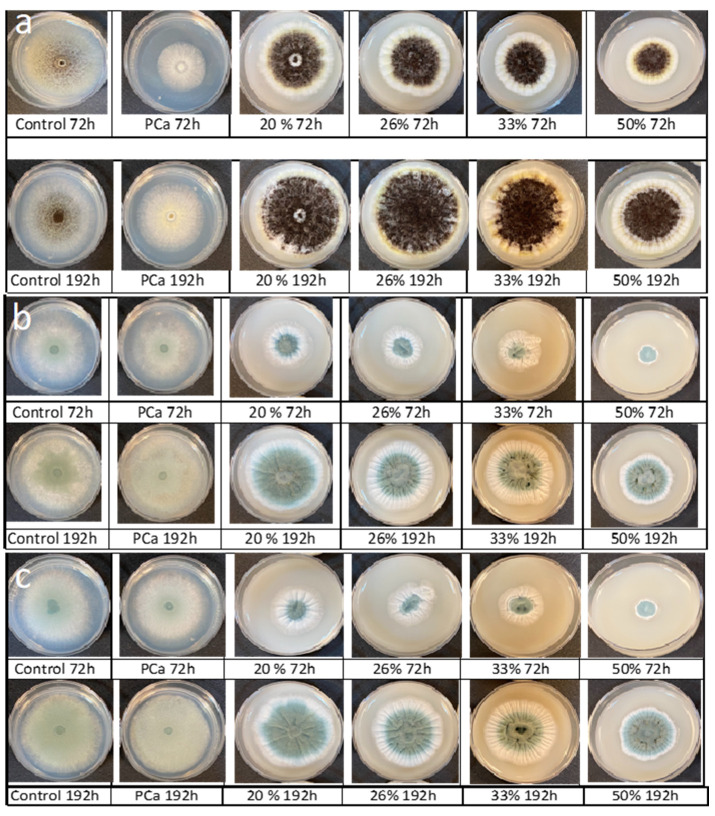
Radial growth of molds treated with different levels of pasteurized aqueous extracts from poolish-type sourdough fermented (48 h) with *Lactiplantibacillus plantarum* NRRL B-4496 (20, 26, 33, or 50%) or 0.3% calcium propionate (PCa) after 72 and 192 h of incubation at 25 °C. (**a**) *Aspergillus niger*, (**b**) *Penicillium chrysogenum*, and (**c**) *Penicillium corylophilum*.

**Figure 6 antibiotics-11-01813-f006:**
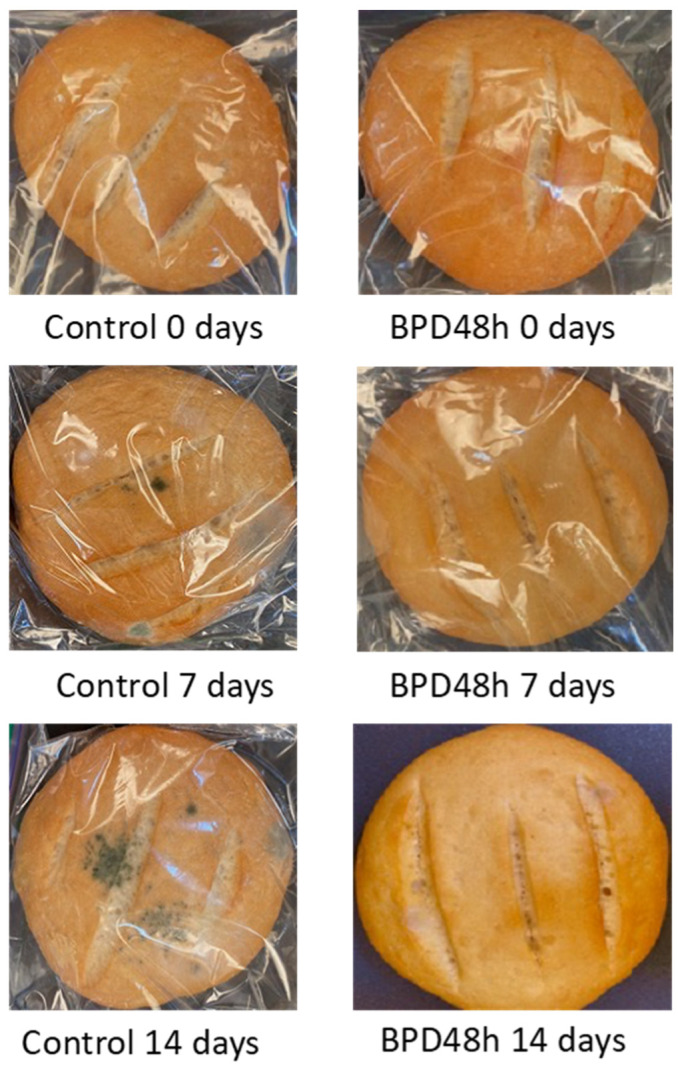
Fungal growth on the control loaves of bread and bread supplemented with fermented poolish-type sourdough with *Lactiplantibacillus plantarum* NRRL B-4496 for 48 h (BPD48h) during storage at 25 °C.

**Figure 7 antibiotics-11-01813-f007:**
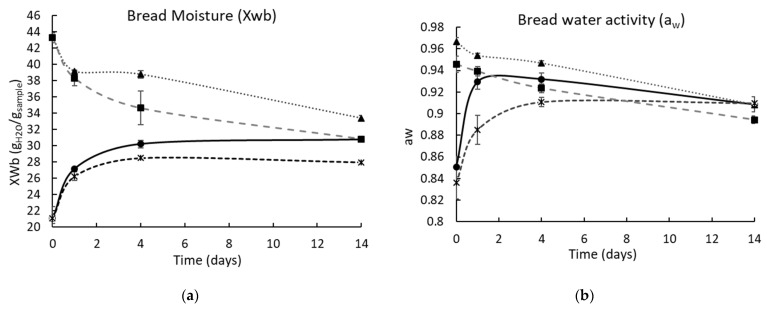
Moisture (**a**) and water activity (**b**) of the bread crust and crumb for 14 days of storage at 25 °C. Control crust (-•-); control crumb (-▲-); BPD48h crust (-*-); and BPD48h crumb (-◾-). BPD48h is bread incorporated with fermented sourdough with *Lactiplantibacillus plantarum* NRRL B-4496 for 48 h.

**Table 1 antibiotics-11-01813-t001:** The pH, titratable acidity, and reducing sugars for different fermented doughs with *Lactiplantibacillus plantarum* NRRL B-4496.

	pH	Titratable Acidity (%)	Reducing Sugars (g/g_100 sample_)
Control	4.62 ± 0.05 a	0.36 ± 0.06 b	0.15 ± 0.01 c
PD 24 h	3.76 ± 0.04 b	1.27 ± 0.09 a	1.48 ± 0.04 b
PD 48 h	3.64 ± 0.01 c	1.40 ± 0.08 a	2.10 ± 0.07 a
PD 72 h	3.63 ± 0.02 c	1.40 ± 0.09 a	2.12 ± 0.20 a

PD: poolish-type dough. Different letters show a significant difference (*p* < 0.05) among samples. Titratable acidity was calculated in weight/volume percentage (%*w*/*v*) as lactic acid.

**Table 2 antibiotics-11-01813-t002:** Parameters (µ maximum growth rate; λ lag phase) of the modified Gompertz equation describing the growth of the molds treated at different concentrations of the pasteurized aqueous extracts from poolish-type sourdough fermented with *Lactiplantibacillus plantarum* NRRL B-4496 for 48 h.

	** *Aspergillus niger* **				
	A	µ (h^−1^)	λ (h)	RMSE	R^2^
Control	2.51 ± 0.018	0.063 ± 0.003	12.06 ± 1.3	0.01	0.995
Control CP	2.41 ± 0.040	0.029 ± 0.003	7.58 ± 3.8	0.03	0.983
20%	2.46 ± 0.026	0.057 ± 0.004	13.3 ± 1.9	0.02	0.991
26%	2.43 ± 0.032	0.055 ± 0.005	13.8 ± 2.5	0.02	0.987
33%	2.17 ± 0.029	0.035 ± 0.003	10.0 ± 2.8	0.02	0.988
50%	2.69 ± 0.028	0.033 ± 0.001	13.6 ± 2.1	0.02	0.995
	** *Penicillium chrysogenum* **				
	A	µ (h^−1^)	λ (h)	RMSE	R^2^
Control	2.62 ± 0.013	0.072 ± 0.003	14.07 ± 0.87	0.01	0.998
Control CP	2.66 ± 0.018	0.076 ± 0.004	16.41 ± 1.30	0.01	0.996
20%	2.51 ± 0.031	0.030 ± 0.002	13.80 ± 2.5	0.02	0.993
26%	2.46 ± 0.035	0.026 ± 0.001	10.30 ± 2.90	0.02	0.992
33%	2.47 ± 0.033	0.028 ± 0.002	17.14 ± 2.65	0.02	0.993
50%	2.50 ± 0.020	0.017 ± 0.005	24.42 ± 2.28	0.01	0.997
	** *Penicillium corylophilum* **				
	A	µ (h^−1^)	λ (h)	RMSE	R^2^
Control	2.63 ± 0.016	0.081 ± 0.004	15.65 ± 1.06	0.01	0.997
Control CP	2.62 ± 0.018	0.079 ± 0.005	16.89 ± 1.30	0.01	0.996
20%	2.50 ± 0.034	0.030 ± 0.002	12.96 ± 2.75	0.02	0.992
26%	2.47 ± 0036	0.025 ± 0.002	12.79 ± 2.88	0.02	0.992
33%	2.45 ± 0.036	0.026 ± 0.002	10.60 ± 2.98	0.02	0.991
50%	2.66 ± 0.022	0.018 ± 0.001	17.20 ± 2.49	0.01	0.996

**Table 3 antibiotics-11-01813-t003:** Lactic and acetic acid concentrations in the aqueous extracts from poolish-type sourdough (PD) fermented with *Lactiplantibacillus plantarum* NRRL B-4496 for 24, 48, or 72 h.

Sourdough	Lactic Acid (mM)	Acetic Acid (mM)
PD24	77.35 ± 5.69 b	19.93 ± 0.33 b
PD48	187.83 ± 4.80 a	35.49 ± 1.97 a
PD72	185.17 ± 1.33 a	36.08 ± 0.96 a

Different letters show a significant difference (*p* < 0.05) among aqueous extracts.

**Table 4 antibiotics-11-01813-t004:** Mold growth on control bread and bread incorporated with fermented polish-type sourdough with *Lactiplantibacillus plantarum* NRRL B-4496.

	% of Breads with Mold Growth	
Time (day)	Control	BPD48h
3	0	0
4	64	0
5	82	0
7	100	0
8	100	0
9	100	0
10	100	50
12	-	50
14	-	50

## Data Availability

The data presented in this study are available on request.

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
