# Peer review of "Antifungal Capacity of Poolish-Type Sourdough Supplemented with Lactiplantibacillus plantarum and Its Aqueous Extracts In Vitro and Bread"

_antibiotics, 2022, doi:10.3390/antibiotics11121813_

Round 1
Reviewer 1 Report
1. The antifungal activity substance need to be identified by kinds of exclusion tests other than directly recognized as organic acid.
2. Lactobacillus plantarum NRRL B-4496 should be assigned to subspecies level; even more, Lactobacillus should be described in accordance with the latest naming convention as Zheng et al. (2020). (Zheng J, Wittouck S, Salvetti E, Franz CMAP, Harris HMB, Mattarelli P, O'Toole PW, Pot B, Vandamme P, Walter J, et al. A taxonomic note on the genus Lactobacillus: Description of 23 novel genera, emended description of the genus Lactobacillus Beijerinck 1901, and union of Lactobacillaceae and Leuconostocaceae. Int J Syst Evol Microbiol 2020; 70:2782-2858.).
Author Response
Review Report (Reviewer 1)
Thank you for taking the time to review our manuscript and for your comments and suggestions.
- The antifungal activity substance need to be identified by kinds of exclusion tests other than directly recognized as organic acid.
We conducted a test, hydrolyzing the extracts with proteinase-K and evaluating the effect of this treatment on the antifungal activity. The suggested changes were made:
Similarly, to determine if the compound responsible was peptide-like, 10 ml of neutralized AE were mixed with 20 mg/ml solution of proteinase K (Sigma-Aldrich, St. Louis, MO, USA) before incubation at 37 °C ± 1 °C for 1 h. Following this, the treated AE was heated further (121°C) for 5 min. Afterward, antifungal activity was evaluated as described before.
The neutralized AEs treated with proteinase K lost their antifungal capacity against all the molds analyzed.
- Lactobacillus plantarum NRRL B-4496 should be assigned to subspecies level; even more, Lactobacillus should be described in accordance with the latest naming convention as Zheng et al. (2020). (Zheng J, Wittouck S, Salvetti E, Franz CMAP, Harris HMB, Mattarelli P, O'Toole PW, Pot B, Vandamme P, Walter J, et al. A taxonomic note on the genus Lactobacillus: Description of 23 novel genera, emended description of the genus Lactobacillus Beijerinck 1901, and union of Lactobacillaceae and Leuconostocaceae. Int J Syst Evol Microbiol 2020; 70:2782-2858.).
The entire document was reviewed, and the suggested changes were made. Throughout the manuscript, Lactobacillus plantarum was changed to Lactiplantibacillus plantarum. The reference of Zheng et al. (2020) was included.
Reviewer 2 Report
Why other organic acids are not studied in the Materials and Methods section besides lactic acid and acetic acid?
Author Response
Review Report (Reviewer 2)
Thank you for taking the time to review our manuscript and for your comments and suggestions.
Why other organic acids are not studied in the Materials and Methods section besides lactic acid and acetic acid?
We included the other acids that were investigated and analyzed in the methodology and results. The following was included:
Standard solutions at 30-400 mM were prepared to quantify lactic, acetic, propionic, butyric, and pyroglutamic acids.
Propionic, butyric, and pyroglutamic acids were not detected in the AE´s
Reviewer 3 Report
The manuscript is an interesting one about developing antibiotics from aqueous extracts from sourdough bread. They found that the lactic acid bacteria obtained inhibited growth of spoilage microorganisms. The photos in Fig. 5 are quite helpful. The results should provide a natural method of bread preservation.
Comments:
Equation 1 has exp 1 in the middle, and that should be checked
Lines 147 and 150: use min and d instead of minutes and days as you did earlier in the paper
Lines 207-7: none AE inhibited should be changed
Line 256: should read "the fit of the prediction model was excellent"
Line 351: H2O and sample should not be subscripts
Author Response
Review Report (Reviewer 3)
Thank you for taking the time to review our manuscript and for your comments and suggestions.
The manuscript is an interesting one about developing antibiotics from aqueous extracts from sourdough bread. They found that the lactic acid bacteria obtained inhibited growth of spoilage microorganisms. The photos in Fig. 5 are quite helpful. The results should provide a natural method of bread preservation.
Comments:
Equation 1 has exp 1 in the middle, and that should be checked
The suggested change was made
We made the change from "exp(1)" to "e," and this was included in the explanation of the model parameters.
Lines 147 and 150: use min and d instead of minutes and days as you did earlier in the paper
The entire document was reviewed, and the suggested changes were made (minutes-min and days-d)
Lines 207-7: none AE inhibited should be changed
The suggested change was made
None AE completely inhibits the growth of studied molds.
Line 256: should read "the fit of the prediction model was excellent"
The suggested change was made
Table 2 shows the parameters of the modified Gompertz equation obtained from mold radial growth; in all cases, the fitted of the prediction model was excellent (R2 > 0.98) (Figure 4).
Line 351: H2O and sample should not be subscripts
The suggested change was made
(~3 gH2O/g sample)
Round 2
Reviewer 1 Report
1. Lactiplantibacillus plantarum only be corrected in abstract, but there was no modification in the text.
2. The written of “p” was wrong.
Author Response
Comments and Suggestions for Authors
Thank you for taking the time to review our manuscript and for your comments and suggestions.
- Lactiplantibacillus plantarum only be corrected in abstract, but there was no modification in the text.
The entire document was reviewed, and the name changes (Lactiplantibacillus plantarum NRRL B-4496) were performed when referring to the strain used. The strains cited from other investigations were left with the original name reported in the original papers.
- The written of “p” was wrong.
The entire document was reviewed, and the change of p (probability) to italics (p) was performed.